# Efficacy and Safety of a Novel Mouthpiece for Esophagogastroduodenoscopy: A Multi-Center, Randomized Study

**DOI:** 10.3390/diagnostics11030538

**Published:** 2021-03-17

**Authors:** Yuichiro Ikebuchi, Kazuya Matsumoto, Naoki Ueda, Taro Yamashita, Hiroki Kurumi, Takumi Onoyama, Yohei Takeda, Akira Yoshida, Koichiro Kawaguchi, Kazuo Yashima, Kazunori Fujiwara, Ryu Imamoto, Hisashi Noma, Masaru Ueki, Hajime Isomoto

**Affiliations:** 1Division of Medicine and Clinical Science, Department of Multidisciplinary Internal Medicine, Faculty of Medicine, Tottori University, Tottori 683-8504, Japan; yama_t11@yahoo.co.jp (T.Y.); kurumi_1022_1107@yahoo.co.jp (H.K.); golf4to@yahoo.co.jp (T.O.); yhytkd7@outlook.jp (Y.T.); akirayoshida1021@yahoo.co.jp (A.Y.); koichiro@tottori-u.ac.jp (K.K.); yashima@tottori-u.ac.jp (K.Y.); 2Irisawa Medical Clinic, Shimane 690-0025, Japan; naoking_16@yahoo.co.jp; 3Department of Gastroenterology, Yasugi Municipal Hospital, Shimane 692-0404, Japan; imaryuu@hotmail.com (R.I.); isomoto@tottori-u.ac.jp (H.I.); 4Department of Otolaryngology, Head and Neck Surgery, Division of Medicine of Sensory and Motor Organs, Faculty of Medicine, Tottori University, Tottori 683-8504, Japan; kfujiwa@tottori-u.ac.jp; 5The Institute of Statistical Mathematics, Tokyo 190-8562, Japan; noma@ism.ac.jp; 6Advanced Medicine, Innovation and Clinical Research Center, Tottori University Hospital, Tottori 683-8504, Japan; masaruueki@gmail.com

**Keywords:** esophagogastroduodenoscopy (EGD), mouthpiece, visual analog scale, gag reflex

## Abstract

This randomized trial aimed to compare the safety and efficacy of the GAGLESS mouthpiece for esophagogastroduodenoscopy (EGD) with that of the conventional mouthpiece. In all, 90 participants were divided into the GAGLESS mouthpiece and conventional mouthpiece groups. The primary endpoint was the severity of pain using the visual analog scale (VAS), and secondary endpoints were examination time, past history of endoscopy, success of the procedure, systolic (SBP) and diastolic (DBP) blood pressure, oxygen saturation, pulse rate before and after EGD, and adverse events. Endoscopy was completed in all cases, and no complications were observed. VAS, when passing the scope through the pharynx, was 2.5 ± 2.4 and 2.0 ± 1.9 cm (*p* = 0.24) in the conventional and GAGLESS groups, respectively, and that, throughout the examination, was 2.5 ± 2.4 and 1.7 ± 1.5 cm (*p* = 0.06), respectively. The difference in blood pressure between the GAGLESS and conventional groups was not significant for SBP (*p* = 0.08) and significant for DBP (*p* = 0.03). The post-EGD difference in DBP was significantly lower in the GAGLESS group than in the conventional group. The results indicate that GAGLESS mouthpieces had a lower VAS during endoscopy than the conventional mouthpieces, and the changes in blood pressure were smaller with the GAGLESS mouthpiece.

## 1. Introduction

Gastric cancer is the third leading cause of cancer-related mortality worldwide [1]. However, the prognosis can be improved by early detection [2], and, if detected early, it can be cured by endoscopic treatment [3]. Therefore, cancer screening programs are important. It has been reported that, within the Korean National Cancer Screening Program, patients who underwent upper endoscopy were less likely to die from gastric cancer [4]. In Japan, Hamashima et al. reported that endoscopic screening could reduce the mortality from gastric cancer by 67% when compared to that with gastric radiographic screening [5]. The challenge with endoscopic screening is the pain associated with endoscopy. In western countries, endoscopy is performed under sedation [6,7]. However, sedation is uncommon in Japan. Although some studies have shown that esophagogastroduodenoscopy (EGD) is more comfortable under sedation. Morbidity and mortality, related to hypoxia due to sedation, have been reported in elderly patients and individuals with underlying cardiopulmonary disease [6,8]. Hence, we developed the GAGLESS mouthpiece (INABA RUBBER, Osaka, Japan) to address the pain associated with endoscopy and have reported the results with this device in the past [9] (Figure 1). In this study, we aimed to compare the safety and effectiveness of the GAGLESS mouthpiece with those of the conventional mouthpiece.

## 2. Materials and Methods

### 2.1. Equipment

The conventional mouthpiece used was the Olympus mouthpiece (MAJ674 mouthpiece, Olympus, Tokyo, Japan), while the GAGLESS mouthpiece was made by INABA RUBBER, Tottori, Japan (Figure 1). All procedures were performed using EGD scopes (GIF-H290 or GIF-HQ290 or GIF-H290Z, Olympus, Tokyo, Japan), and all EGD examinations were performed under air insufflation. EGD was performed by physicians with more than six years of experience in endoscopy.

### 2.2. Patients and Study Design

All patients provided written informed consent before the EGD. Between October 2018 and March 2019, all consecutive patients undergoing EGD at the Tottori University Hospital or Yasugi Municipal Hospital were screened. The inclusion criterion was males aged above 20 years (because the GAGLESS mouthpiece was created exclusively for males). All patients were informed about the aims, methods, and possible adverse effects of the procedure, and signed written consent was obtained from all patients. In all, 90 participants were recruited and divided into two equal-sized groups (the conventional mouthpiece group [Conventional group] and the GAGLESS mouthpiece group [GAGLESS group]) using a randomized number table prepared in advance by a researcher who was not involved in the study. The study was approved by the Institutional Review Board of the Tottori University Hospital (IRB No. #18B015).

### 2.3. Allocation Method

Random allocation was performed by the stratified replacement block method with “facility” as the layer. A statistician created a randomized allocation table using SAS Ver. 9.4 (SAS Institute Inc., Cary, NC, USA), and each facility allocated the patients based on this table. This was an open-label study, and neither the participants nor the research staff were masked for the allocation. However, the stratified block size was known only by the statistician responsible for creating the allocation table and was masked for the research staff involved in patient recruitment. The allocation ratio was 1:1.

### 2.4. Outcome Assessment and Evaluation

The primary endpoint of the study was the severity of pain, as recorded on a 10 cm visual analog scale (VAS) after EGD. The scores on the 10 cm VAS indicated ranges from “no pain” on the left to “pain as bad as it could be” on the right side. Two VAS scores were recorded when passing the endoscope through the pharynx (VAS1) and throughout the examination (VAS2). The secondary endpoints measured were examination time, history of past endoscopy, success of the procedure, and comparison of systolic blood pressure (SBP), diastolic blood pressure (DBP), oxygen saturation of the peripheral artery (SpO2), pulse rate (PR) before and after EGD, and adverse events.

### 2.5. Statistical Analysis

Continuous data are summarized as means and standard deviations, and categorical data are summarized as numbers (proportions). For analyses of the primary endpoint, the VAS scores were compared using Welch’s *t*-test. Additionally, the examination time, history of past endoscopy, the success rate of the procedure, ease of insertion by the examiner, and accidents were compared using the Welch’s *t*-test for continuous variables or Fisher’s exact test for categorical variables. These analyses were performed based on the intention-to-treat principle.

Data from the pilot study showed that the mean VAS score with the conventional mouthpiece was 4.27 cm (standard deviation ± 1.76), VAS score with the GAGLESS mouthpiece was 1.98 cm (standard deviation ± 2.37), and the difference in the VAS score between the groups was 2.29 cm.The sample size was estimated conservatively based on the differences in the mean values between the groups and was set to 1.83 cm. The standard deviation was set to 2.07 cm, and the sample size required to achieve a detection power of 95% was calculated. As a result, 35 cases per group were required. Moreover, considering the uncertainty and probable errors from the data of the pilot test, we set the sample size to 45 cases per group.

All statistical tests were two-sided with a significance level of 0.05 and were performed using SAS version 9.4 (SAS Institute).

## 3. Results

A total of 90 patients underwent EGD. Forty-five people each were assigned to the conventional group and the GAGLESS group by stratified randomization. Patient characteristics are shown in Table 1. Both groups were well balanced in terms of age, history of previous endoscopic examination, SBP, DBP, PR, and SpO2.

The results are shown in Table 2. Endoscopy was completed in all cases, and no complications were observed in either group. The average procedure time was 367 ± 220 s and 358 ± 129 s in the conventional and GAGLESS groups, respectively, and there was no significant difference between the groups. The VAS1 during the pharyngeal passage of the endoscope was 2.5 ± 2.4 cm and 2.0 ± 1.9 cm (*p* = 0.24) in the conventional and GAGLESS groups, respectively, and the VAS2 throughout the endoscopic examination was 2.5 ± 2.4 cm and 1.7 ± 1.5 cm (*p* = 0.06), respectively. The GAGLESS group had a lower VAS score compared to the conventional group, even though the differences were not significant.

Results of the analyses of secondary outcomes are also shown in Table 2. In the secondary endpoints, there was no significant difference in the SpO2 and pulse before and after the endoscopic examination between the groups. However, the *p* value for changes in the blood pressure difference between the conventional and GAGLESS groups was *p* = 0.08 for SBP and *p* = 0.03 for DBP. The change in DBP was significantly lower in the GAGLESS group than in the conventional group (Figure 2). No adverse events were observed throughout this study.

## 4. Discussion

We previously reported that GAGLESS mouthpieces could suppress the gag reflexes by attaching firmly to the back teeth [9]. In the present report, when the scope passed through the pharynx, the overall VAS was lower with the GAGLESS mouthpiece than with the conventional mouthpieces. Additionally, the difference in DBP before and after the endoscopic examination was significantly small with the GAGLESS mouthpiece, suggesting that it might be less likely to cause a change in the hemodynamics compared to that with the conventional mouthpieces.

It is expected that the image accuracy of EGD will improve consistently in the coming years and new technologies such as ultra-magnifying endoscopes will be developed. Thus, the duration of endoscopic examination might increase [10,11]. Meanwhile, methods for reducing the pain associated with endoscopic examination have not yet been fully developed. Sedated endoscopy has various risks such as a decrease in BP, aggravation of the respiratory condition, longer examination time, and higher cost [7]. Thin endoscopy, such as transnasal endoscopy, has been developed and is expected to reduce the problem of pain associated with transoral endoscopy. However, thin endoscopy has complications including nasal pain, epistaxis, long examination time, and deterioration of the image quality [12,13]. Additionally, VAS2 of 2.3–3.9 cm was reported using transnasal endoscopy in a previous study [14]. In our study, the VAS2 of the GAGLESS mouthpiece was as low as 1.7 ± 1.5 cm, which is better than that reported in previous studies using trans-nasal endoscopy. Sedation was also not used in this study. Un-sedated GAGLESS mouthpiece endoscopy is well-tolerated, feasible, and safe for patients. It is similar to trans-nasal endoscopy without sedation. The VAS2 was lower than the VAS1 in the GAGLESS group. However, there was no change in the conventional group. In sports, a mouthpiece that attaches with the hind teeth is actively adopted, and the cardiopulmonary ability is reported to improve [15,16]. It is possible that the VAS2 was lower than the VAS1 because the pain was maximized when passing the endoscope through the pharynx, and the GAGLESS mouthpiece then provided a relaxing effect.

In this study, the GAGLESS group had a significantly lower DBP than the conventional group. In general, blood pressure and heart rate increase after endoscopy [17]. This might be due to the increased sympathetic activity caused by mechanical stimulation of the pharynx by endoscopy, anxiety due to the examination, and mental stress, which all lead to an increase in blood pressure [18]. The GAGLESS mouthpiece has a U-shape that follows the entire dental arch. It is made of soft material so that it can be comfortably pinched when chewing with the teeth. Moreover, it provides occlusal stability when chewing with the hind teeth. We believe that the GAGLESS mouthpiece causes lesser sympathetic activation and is less stressful for the cardiovascular system.

There are several limitations in this report. First, there were few patients with no past history of endoscopy. Since the number of patients undergoing endoscopy for the first time in this study was small, the VAS score might have been lower in both groups than in the past reports. The low overall VAS score possibly failed to show a significant difference in the outcomes. Hence, we plan to recruit only patients undergoing a first-time endoscopy in the future. Next, the number of cases included in this study was small. The VAS score of the conventional group was also lower than that reported in the previous studies. Therefore, it is difficult to identify a significant difference. This indicates that it is necessary to increase the number of patients in future studies.

The severe acute respiratory syndrome coronavirus 2 (SARS-CoV-2) that causes the coronavirus disease (COVID-19) also affected endoscopic examinations [19]. COVID-19 is mainly transmitted through aerosols or by direct contact [20,21]. Endoscopy can induce vomiting and increases the risk of transmitting the SARS-CoV-2 infection. We have previously reported that GAGLESS mouthpieces suppress the vomiting reflex [9]. Thus, it can reduce the risk of transmission of COVID-19 during endoscopic examinations.

## 5. Conclusions

In this study, all endoscopic examinations in the GAGLESS mouthpiece group could be safely performed. Participants using the GAGLESS mouthpiece had a lower VAS during endoscopy than those using conventional mouthpieces, and changes in the blood pressure were also small in the GAGLESS group, suggesting that the GAGLESS mouthpiece would be effective in reducing pain associated with EGD.

## Figures and Tables

**Figure 1 diagnostics-11-00538-f001:**
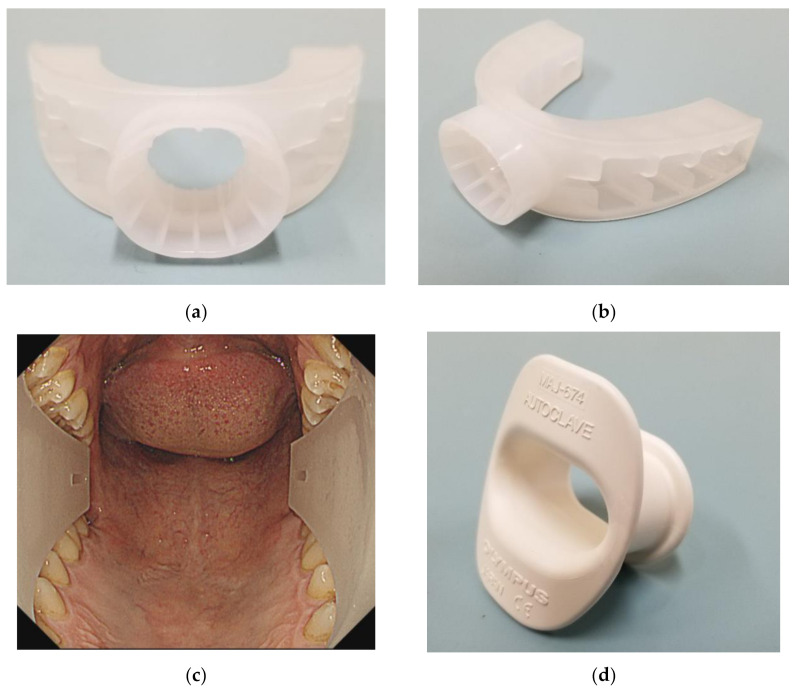
GAGLESS mouthpiece. (**a**) Front view. (**b**) Actual oblique view. (**c**) Oropharyngeal view with the endoscope. GAGLESS mouthpiece was U-shaped along the entire dental arch. (**d**) Conventional mouthpiece and actual oblique view. The conventional mouthpiece is held primarily by the front teeth.

**Figure 2 diagnostics-11-00538-f002:**
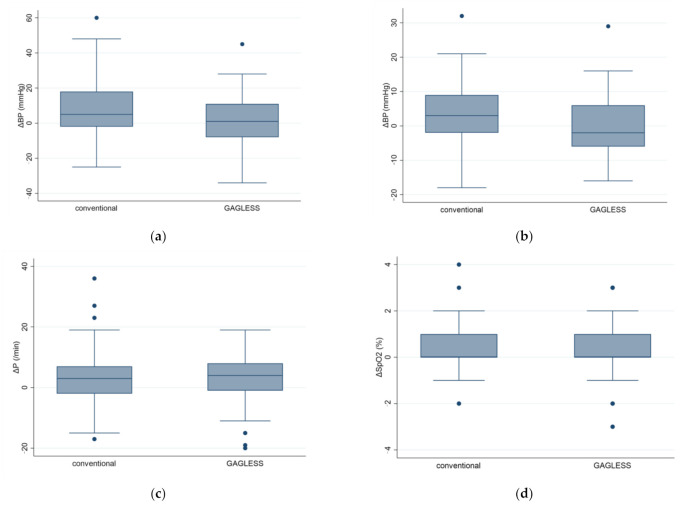
Cardiovascular responses in the GAGLESS group and the conventional group. Changes in (**a**) systolic blood pressure (SBP), (**b**) diastolic blood pressure (DBP), (**c**) pulse rate (PR), and (**d**) oxygen saturation by pulse oximetry (SpO2) were evaluated before and after scope insertion. Values represent means and standard errors of the mean. * *p* < 0.05.

**Table 1 diagnostics-11-00538-t001:** Patient characteristics.

	Conventional Group *N* = 45	GAGLESS Group *N* = 45	*p*-Value
Age, Mean ± SD	62.3 ± 11.9	64.5 ± 12.4	0.38
First Time, (%) *	2 (4)	0 (0)	0.49
SBP, mmHg	127.8 ± 23.3	129.9 ± 19.2	0.64
DBP, mmHg	71.4 ± 15.3	71.2 ± 13.2	0.95
PR, /min	72.5 ± 12.1	76.7 ± 18.0	0.19
SpO2, %	97.2 ± 1.4	96.9 ± 1.8	0.48

SBP: systolic blood pressure, DBP: diastolic blood pressure, PR: pulse rate, SpO2: saturation of peripheral artery. * First time: history of endoscopy examination.

**Table 2 diagnostics-11-00538-t002:** Results of the primary and secondary endpoints.

	Conventional Group *N* = 45	GAGLESS Group *N* = 45	*p*-Value
Success Rate, %	100	100	-
Adverse Events	None	None	-
Procedure Time, Seconds	366.8 ± 220	358.3 ± 130	0.82
VAS1 *, cm	2.5 ± 2.4	2.0 ± 1.9	0.24
VAS2 **, cm	2.5 ± 2.4	1.7 ± 1.5	0.06
Differences in SBP, mmHg	8.0 ± 15.4	2.1 ± 15.8	0.08
Differences in DBP, mmHg	4.3 ± 9.7	−0.1 ± 9.2	0.03
Differences in PR, /min	3.3 ± 10.3	2.8 ± 8.8	0.83
Differences in SpO2, %	0.22 ± 1.1	0.28 ± 1.5	0.81

VAS: visual analog scale, SBP: systolic blood pressure, DBP: diastolic blood pressure, PR: pulse rate. * VAS1: when the endoscope passes through the pharynx. ** VAS2: throughout the endoscopic examination.

## Data Availability

Data available in a publicly accessible repository. The data presented in this study are openly available in [repository name e.g., FigShare] at [doi], reference number [reference number].

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
