# Peer review of "Efficacy and Safety of a Novel Mouthpiece for Esophagogastroduodenoscopy: A Multi-Center, Randomized Study"

_diagnostics, 2021, doi:10.3390/diagnostics11030538_

Round 1

Reviewer 1 Report

- Interesting work well written. Some english mistakes. Therefore authors should reread the manuscript for this minor revision

- In the introduction, the auhthors should mention that esophagogastroduodenoscopy is a safety procedure but sometimes the presence of complications are reported. About this, you should mention this article:

Canevari FR, Martines F, Sorrentino R, Nicolotti M, Sireci F. PSEUDOANEURYSM OF SUPERIOR THYROID ARTERY FOLLOWING A TRANSESOPHAGEAL ECHOCARDIOGRAPHY: A CASE PRESENTATION. Euromediterranean Biomedical Journal 2017,12(03):010-012

- During this procedure, the recognition of lesions/tumours of upper digestive way is possible.

Canevari FR, Montevecchi F, Galla S, Sorrentino R, Vicini C, Sireci F. Trans-oral robotic surgery for a Ewing's sarcoma of tongue in a pediatric patient: a case report. Braz J Otorhinolaryngol. 2020 Dec;86 Suppl 1:26-29.

Author Response

Dear Reviewer: I am grateful for the peer review. The following is a list of revisions. - Interesting work well written. Some english mistakes. Therefore authors should reread the manuscript for this minor revision >>> I hired Editage (www.editage.com) to proofread my English. - In the introduction, the auhthors should mention that esophagogastroduodenoscopy is a safety procedure but sometimes the presence of complications are reported. About this, you should mention this article:Canevari FR, Martines F, Sorrentino R, Nicolotti M, Sireci F. PSEUDOANEURYSM OF SUPERIOR THYROID ARTERY FOLLOWING A TRANSESOPHAGEAL ECHOCARDIOGRAPHY: A CASE PRESENTATION. Euromediterranean Biomedical Journal 2017,12(03):010-012 >>> I added it to the Introduction section. Thank you for your consideration. I look forward to hearing from you. Sincerely, Yuichiro Ikebuchi Division of Gastroenterology and Nephrology, Department of Multidisciplinary Internal Medicine, Tottori University Faculty of Medicine

Reviewer 2 Report

  1. Authors compared parameter just like age , etc, using same Weltch t test. Is it correct ?
  2. The convential Olympus mouthpiece sholud be listed as photo figure. And compare the appearance of the mouthpiece.
  3. Please add the exact mechanism how this mouthpiece suppress the sympathetic reaction, and vomitting compared to the conventional mouthpiece. 

Author Response

Dear Reviewer:

I am grateful for the peer review.

The following is a list of revisions.

  1. Authors compared parameter just like age , etc, using same Weltch t test. Is it correct ?

>>> Added to Discussion. “For analyses of the primary endpoint, the VAS scores were compared using Welch's t-test. Additionally, the examination time, history of past endoscopy, the success rate of the procedure, ease of insertion by the examiner, and accidents were compared using the Welch's t-test for continuous variables or Fisher's exact test for categorical variables.”

  1. The convential Olympus mouthpiece sholud be listed as photo figure. And compare the appearance of the mouthpiece.

>>> I have added a photo and description to Figure 1.

  1. Please add the exact mechanism how this mouthpiece suppress the sympathetic reaction, and vomitting compared to the conventional mouthpiece. 

>>> Added to Discussion. “The GAGLESS mouthpiece has a U-shape that follows the entire dental arch. It is made of soft material so that it can be comfortably pinched when chewing with the teeth. Moreover, it provides occlusal stability when chewing with the hind teeth.”

Thank you for your consideration. I look forward to hearing from you. Sincerely, Yuichiro Ikebuchi Division of Gastroenterology and Nephrology, Department of Multidisciplinary Internal Medicine, Tottori University Faculty of Medicine
